# Tuning the Hydrophobicity of a Hydrogel Using Self-Assembled Domains of Polymer Cross-Linkers

**DOI:** 10.3390/ma12101635

**Published:** 2019-05-19

**Authors:** Hee-Jin Kim, Sungwoo Cho, Seung Joo Oh, Sung Gyu Shin, Hee Wook Ryu, Jae Hyun Jeong

**Affiliations:** Department of Chemical Engineering, Soongsil University, 369, Sangdo-Ro, Dongjak-Gu, Seoul 06978, Korea; kimhj0706@ssu.ac.kr (H.-J.K.); som113@ssu.ac.kr (S.C.); ohsj0610@ssu.ac.kr (S.J.O.); whitegd45@ssu.ac.kr (S.G.S.); hwryu@ssu.ac.kr (H.W.R.)

**Keywords:** hydrogel, hydrophobicity, self-assembly, degree of swelling

## Abstract

Hydrogels incorporated with hydrophobic motifs have received considerable attention to recapitulate the cellular microenvironments, specifically for the bio-mineralization of a 3D matrix. Introduction of hydrophobic molecules into a hydrogel often results in irregular arrangement of the motifs, and further phase separation of hydrophobic domains, but limited efforts have been made to resolve this challenge in developing the hydrophobically-modified hydrogel. Therefore, this study presents an advanced integrative strategy to incorporate hydrophobic domains regularly in a hydrogel using self-assembled domains formed with polymer cross-linkers, building blocks of a hydrogel. Self-assemblies formed by polymer cross-linkers were examined as micro-domains to incorporate hydrophobic motifs in a hydrogel. The self-assembled structures in a pre-gelled solution were confirmed with the fluorescence analysis and the hydrophobicity of a hydrogel could be tuned by incorporating the hydrophobic chains in a controlled manner. Overall, the results of this study would greatly serve to tuning performance of a wide array of hydrophobically-modified hydrogels in drug delivery, cell therapies and tissue engineering.

## 1. Introduction

Hydrogels have been extensively studied for use in various biomedical applications including drug delivery, tissue engineering and recently, Bio-MEMS (bio-microelectromechanical system) [1,2,3,4]. The successful use of the hydrogels in these applications greatly relies on their physical and chemical properties for maximizing their functionality [5,6,7,8,9]. Recently, mineralized hydrogel systems are being increasingly studied to understand bio-mineralization processes related to the development, repair, regeneration and remodeling of bone tissue [10,11]. In here, the hydrogels incorporated with hydrophobic motifs are required to recapitulate the hydrophobic/hydrophilic microenvironments in mineralized bone tissue [12,13]. Extensive efforts have been made to control the hydrophobicity of hydrogels [13,14,15], and good examples include latex particles with coated hydrophobic segments to enhance the performance of hydrogels [15]. However, incorporating of hydrophobic chains into a hydrogel often results in irregular arrangement of the hydrophobic chains, and further phase separation into hydrophobic domains, but limited efforts have been made to resolve this challenge in the hydrophobically-modified hydrogel.

Therefore, we hypothesized that self-assemblies of polymer cross-linkers in a pre-gelled solution would allow us to incorporate hydrophobic chains regularly as micro-domains in a hydrogel. A polymer cross-linker, building block of a hydrogel has two or more acrylate groups, which are relatively hydrophobic, compared to hydrophilic backbone polymer. Then, the polymer cross-linkers would be self-assembled with hydrophobic cores, which can be used as micro-domains for incorporating hydrophobic molecules regularly (Figure 1b). This hypothesis was examined using a model system for a hydrophobically-modified hydrogel formed by cross-linking of poly(ethylene glycol) diacrylate (PEGDA). The self-assembled structures of PEGDAs in a pre-gelled solution were confirmed with the fluorescence analysis (Figure 1b). Then, poly(propylene glycol) methacrylate (PPGMA) with varying of mass fraction was used as model hydrophobic chains (Figure 1c). The effects of the hydrophobic domains incorporated into a hydrogel were studied by measuring swelling ratio and contact angle of a hydrogel. The underlying mechanism by which micro-domains provided by self-assembling of polymer cross-linkers tuned the hydrophobicity in a hydrogel was examined by evaluating average pore size of a hydrogel and characterizing by fluorescence analysis. Overall, this study demonstrates a novel strategy to create a hydrogel incorporated with hydrophobic chains in a controlled manner by self-assembling of polymer cross-linkers.

## 2. Materials and Methods

### 2.1. Fluorescent Analysis of Self-Assembling of Polymer Cross-Linkers in a Solution

The self-assemblies of polymer cross-linkers in a pre-gelled solution were investigated using a pyrene probe. [14,15]. The pyrene (Sigma, Peabody, MA, USA) was dissolved in acetone to prepare a stock solution with a concentration of 6.0 × 10^−4^ M. Polymer cross-linker pre-gelled solutions were prepared in DI (deionized) water (2 mL) by poly(ethylene glycol) diacrylate of M_n_ 575 g/mol (PEGDA-575, Sigma) and M_w_ 3400 g/mol (PEGDA-3400, Sigma), respectively. In parallel, acrylate group-free poly(ethylene glycol) of M_n_ 3350 g/mol (PEGdiol, Sigma) were dissolved in DI water as a control. Then, the pyrene solution was dropped into the polymer solutions with varying polymer concentrations. The mixture of the polymer solution and pyrene was sonicated for 10 min to ensure dispersion of pyrene in the polymer solution. The mixture was further incubated at room temperature for at least 12 h in the dark, so the pyrene was preferentially associated with hydrophobic domains of polymers. The mixture loaded in a quartz cuvette was excited at a wavelength of 330 nm and a resulting emission spectrum was obtained using photo luminescence (QM40, Photon Technology International, HORIBA, Japan). The band-width was adjusted to 2.0 nm for both excitation and emission.

### 2.2. Hydrogel Preparation

Pre-gelled solutions were prepared by mixing 10 wt % PEGDA-575, which has a hydrophilic property and poly(propylene glycol) methacrylate (PPGMA, M_n_ of 375 g/mol, Sigma), acting hydrophobic chain, by increasing of mass fraction under fixed total polymer concentration, 10 wt %. The PEGDA was dissolved in DI water at 40 wt % stock solution, and PPGMA was dissolved in dimethyl sulfoxide (DMSO, Sigma, Peabody, MA, USA) at 20 wt % stock solution. The photo-initiator, 2-hydroxy-4′-(2-hydroxyethoxy)-2-methylpropiophenone (Irgacure 2959, Sigma) was dissolved in DMSO at 10 wt % stock solution and added to the 1 mL pre-gelled solution to form 0.2 wt % as the final concentration. First, the 1 mL of pre-gelled solution mixed by vortex mixer. Second the mixed pre-gelled solution was cast between glass plates with spacer of 1 mm of thickness. Then, the cast pre-gelled solution was exposed by UV lamp (365 nm, VL-4.LC, VILBER LOURMAT, France) for gelation about 10 min, after then, the gel was punched with 8 mm diameter. Gel disks were immersed in DI water to remove unreacted hydrophilic residuals and then, subsequently immersed in DMSO to residual hydrophobic polymers in the cross-linked gel disk. Finally, gel disk was immerged to exchange DMSO to DI water and further incubated in DI water for 24 h before characterizations described below.

### 2.3. Hydrogel Characterization

The swelling ratio of the hydrogel at equilibrium was determined by measuring the weight of the hydrated gel and that of the dried gel. The hydrogels were immersed in DI water at room temperature and for 24 h to ensure the hydrogels were fully hydrated [16]. The degree of swelling (*Q*), defined as the reciprocal of the volume fraction of a polymer in a hydrogel (*v_2_*), was calculated from the following Equation (1),
(1)$$Q=\nu_{2}^{-1}=\rho_{p}\left[\frac{Q_{m}}{\rho_{s}}+\frac{1}{\rho_{p}}\right]$$
where *ρ_s_* is the density of water, *ρ_p_* is the density of polymer and *Q_m_* is the swelling ratio, the mass ratio of swelled gel to the dried gel.

The average pore size (ξ) of hydrogel was calculated from the polymer volume fraction (*v*_2,s_) and the unperturbed mean-square end-to-end distance of the monomer unit (*r_o_*^−2^) using Equations (2) and (3):(2)$$\xi =\left(\nu_{2,s}^{-1/3}\right){\left(r_{0}^{-2}\right)}^{1/2}$$
(3)$$\left(r_{0}^{-2}\right)=l{\left(2\frac{{\bar{M}}_{c}}{{\bar{M}}_{r}}\right)}^{1/2}\;C^{1/2}=l{\left(2n\right)}^{1/2}C^{1/2}$$
where l is the average value of the bond length between C–C and C–O bonds in the repeatable unit of PEG [–O–CH_2_–CH_2_–], which is taken as 1.46 Å; *M_c_* is the average molecular mass between cross-links in the network; *M_r_* is the molecular mass of the PEG repeating unit; *n* is the number of repeat unit, which is taken as 7 (PEGDA M_n_ of 575 g/mol) and C is the characteristic ratio for poly(ethylene glycol), which is taken here as 4 [17].

The inner micro-structures of the hydrophobically-modified hydrogels were analyzed by SEM (FE-Scanning electron microscope, JEOL-7001F, JEOL Ltd., Tokyo, Japan) with an acceleration voltage of 5 kV. The hydrogels were lyophilized and covered with a platinum (Pt) layer produced by a vapor deposition. Values of the water contact angle (*θ_w_*) on the surfaces of the hydrogels were measured by depositing a drop of water (4.0 μL) under atmospheric condition with DSA100 (KRÜSS, Hamburg, Germany). The contact angle was measured immediately after placing the water drop on the hydrogel surface [18]. Bovine serum albumin (BSA, Sigma) was also used as a model protein to evaluate the protein release rate from the hydrophobically-modified hydrogels [19]. The PEGDA and PPGMA stock solution were dissolved in DI water to prepare the pre-gelled solution at 10 wt %. The pre-gelled solution was mixed with BSA and photo-initiator, the mixture immediately cast between glass plates with spacers of 1 mm thickness, and exposed by UV lamp about 10 min. After 10 min, gel disks with a diameter of 8 mm were punched. Each hydrogels containing BSA was suspended in 400 μL of Dulbecco’s phosphate buffered saline (PBS, biowest) and incubated at 37 °C. At a designated time point, each PBS immersed hydrogel containing BSA was removed. Then, hydrogels were filled with fresh PBS again thereby keeping the sink condition. BSA amount released from the hydrogel was quantitatively measured using Pierce^TM^ BCA protein assay kit (Thermo Scientific, USA) according to the manufacturer’s instructions. The absorbance was measured at 562 nm using ELISA (Multiskan GO, Thermo Scientific, USA). The BSA release profile obtained was fitted to the Ritger-Peppas equation [20].
(4)$$\frac{M_{t}}{M_{\infty }}=k\cdot t^{n}$$
where *M_t_* is the cumulative amount of protein released at the time, t; *M**_∞_* is the total amount of protein in the hydrogels; k is the kinetic rate constant and *n* is the exponent related to the release mechanism.

## 3. Results and Discussion

### 3.1. Analysis of Self-Assembling of Polymer Cross-Linkers in a Solution

This study presents an effective method to incorporate hydrophobic domains regularly in a hydrogel by self-assembling of polymer cross-linkers. First, the self-assembled structures formed in a pre-gelled solution were confirmed with the fluorescence analysis. The hydrophobic association between the polymer cross-linkers with acrylate groups that are slightly hydrophobic was examined with a ratio of the third-to-first vibrational fine structure (*I_3_*/*I_1_*) in the fluorescence spectrum of pyrene prove (Figure 2). Generally, the *I_1_* peak arises from the transition that can be enhanced by the distortion of the π-electron cloud [21,22]. Therefore, as the microenvironment of pyrene becomes more polar, the *I_1_* peak becomes more notable at the expense of other peaks (*I_3_*). That means the ratio of *I_3_*/*I_1_* represents the degree of self-assemblies between acrylate groups linked to the polymer cross-linkers. *I_3_/I_1_* of the PEGDA polymer cross-linker solutions increased as the PEGDA concentration exceeded a critical concentration (critical aggregation concentration, CAC), which means the self-assemblies are formed at this point, as shown in Figure 2. In contrast, *I_3_/I_1_* of the PEGdiol polymer without acrylate groups was independent of PEGdiol concentration. The *I_3_/I_1_* of the PEGdiol solution was approximately 0.5, which is characteristic value for pyrene dispersed in water. Therefore, this fluorescent analysis demonstrated that polymer cross-linkers with acrylate groups are self-assembled in an aqueous solution because of the hydrophobic association between acrylate groups [15].

### 3.2. Effects of the Hydrophobic Domains Incorporated into a Hydrogel

The hydrophobically-modified hydrogels were prepared via in situ radical polymerization of self-assembled PEGDAs and PPGMAs with varying mass fraction. As pointed out above, the concentration of polymer cross-linkers used to form a hydrogel was higher than the CACs of cross-linkers as shown in Figure 2c. Figure 3 showed that in general, the degree of swelling (*Q*) of hydrogels increased as the concentration of pure PEGDAs decreased due to the increase of average pore-size (ξ) of hydrogel. For example, 2.31 nm of ξ in 10 wt % of PEGDAs increases to 2.48 nm of that in 8.0 wt % of PEGDAs as shown in Table 1, calculated using the Equation (1). However, interestingly, in the case of the hydrophobically-modified hydrogels, the *Q* of hydrogel decreased with increasing PPGMA portion from 0 wt % to 0.5 wt %, despite of the decrease of PEGDA concentration and then, the *Q* was increased with increasing PPGMA from 0.5 wt % to 2.0 wt %. These results indicated that PPGMA regulates the degree of swelling of the hydrogel at a mass fraction of less than 0.5 wt % of PPGMA as a hydrophobic repulsion. In contrast, above 0.5 wt % of the mass fraction of PPGMAs, the decreased mass fraction of PEGDAs had a predominant role to the *Q* of hydrogel. Note that PPGMAs are not related to regulate the ξ of hydrogel, because the PPGMA molecule has a single acrylate group, which is not acting as a cross-linker. However, the ξ of hydrogel from 0 wt % to 0.5 wt % of PPGMAs was decreased despite of the decrease of PEGDAs’ concentration. As addressed above, this result is attributed to hydrophobic repulsion by PPGMA.

The inner micro-structures of hydrophobically-modified hydrogels (HMHs) were examined with freeze-dried gels using SEM. Figure 4 shows the images of hydrogels depending on the introduced amounts of PPGMAs. Incorporation of the hydrophobic PPGMAs into the hydrogel made minimal difference in the 3D networked microstructures as compared to the pure PEGDA hydrogel (HMH-1; Figure 4a,b). Therefore, the pore-size of the hydrogel was minimally affected by incorporating the hydrophobic chains. However, the hydrogel with 2.0 wt % of PPGMAs (HMH-4) exhibited relatively large hydrophobic micro-domains (Figure 4c-1,c-2). This result implicates that hydrophilic PEGDA and hydrophobic PPGMA molecules were separated into a two phases in the pre-gelled solution. As confirmed by the fluorescence assay, the PEGDAs, building blocks of a hydrogel are self-assembled with hydrophobic cores, which can be used as micro-domains for incorporating PPGMA. However, incorporating of PPGMAs over a certain amount that is likely the loading capacity of the self-assemblies of PEGDAs resulted in the phase separation. Accordingly, the water contact angles (*θ_w_*) measured at the surface of hydrogel increased with increasing of incorporating PPGMAs but, the increase in the water contact angle has limited in HMH-4, due to the irregular arrangements of hydrophobic chains (Figure 4c-3). As a result, the hydrophobicity of a hydrogel could be tuned in a controlled manner by incorporating hydrophobic molecules using self-assembled domains of polymer cross-linkers.

BSA was encapsulated into the HMHs using PEGDA-3400 to evaluate the effects of the hydrophobic motifs on the protein release rate (Table 2). The release rate of BSA from the HMH gel was quantified by measuring the amount of BSA released into the incubation media on a daily basis over ten days (Figure 5). Increasing the hydrophobicity from 0 wt % to 2 wt % of PPGDAs significantly decreased the amount of BSA initially released from the HMH gels (Figure 5a). The cumulative mass fraction of BSA released from HMHs over 12 h, M_t_/M_∞_, was fitted with the Ritger-Peppas equation to calculate a kinetic rate constant (*k*; Figure 5b). The *k* of the HMHs decreased by 30% as PPGMAs was increased from 0 wt % to 2 wt %. The hydrogels incorporated with hydrophobic motifs would be useful to study for releasing and secreting of soluble and insoluble factors in the cellular microenvironment. In summary, the self-assemblies formed with polymer cross-linkers in a pre-gelled solution could be a place of incorporation of hydrophobic motifs regularly and stably in thermodynamics upon the critical range. However, above this critical point, the self-assemblies would not provide enough room for the hydrophobic parts, so that the phase will be separated and consequently irregular hydrophobic macro-domains in size and arrangement will be generated.

## 4. Conclusions

Taken together, this study presents a new strategy to incorporate hydrophobic domains regularly in a hydrogel by self-assembling of polymer cross-linkers, building blocks of a hydrogel. Self-assemblies between polymer cross-linkers were examined as micro-domains to incorporate hydrophobic motifs in a hydrogel. The self-assembled structures in a pre-gelled solution were confirmed with the fluorescence analysis and the hydrophobicity of a hydrogel could be tuned by incorporating the motifs in a controlled manner. Overall, the results of this study would greatly serve to tuning performance of a wide array of hydrophobically-modified hydrogels in drug delivery, cell therapies and tissue engineering.

## Figures and Tables

**Figure 1 materials-12-01635-f001:**
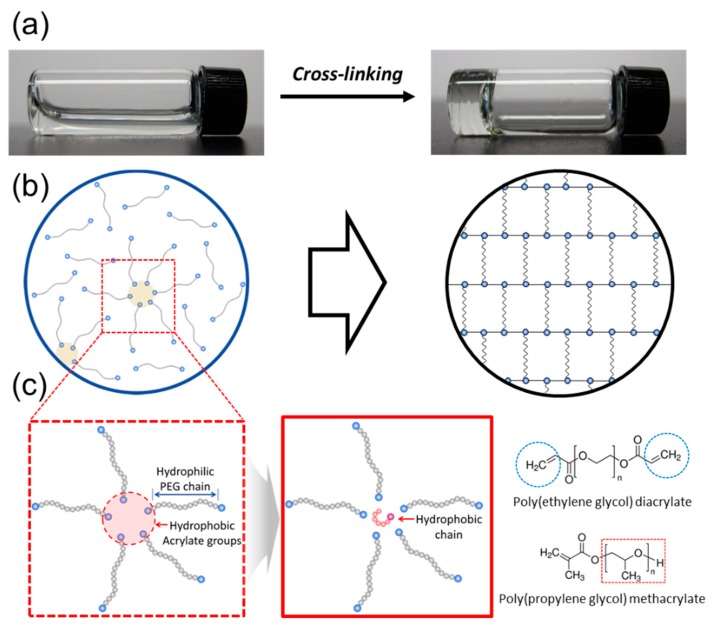
Schematic description of hydrogel forming from polymer cross-linkers in a pre-gelled solution (**a**). The internal structures of self-assemblies associated with polymer cross-linkers in a pre-gelled solution (**b**) and incorporating of hydrophobic chains using the self-assemblies (**c**).

**Figure 2 materials-12-01635-f002:**
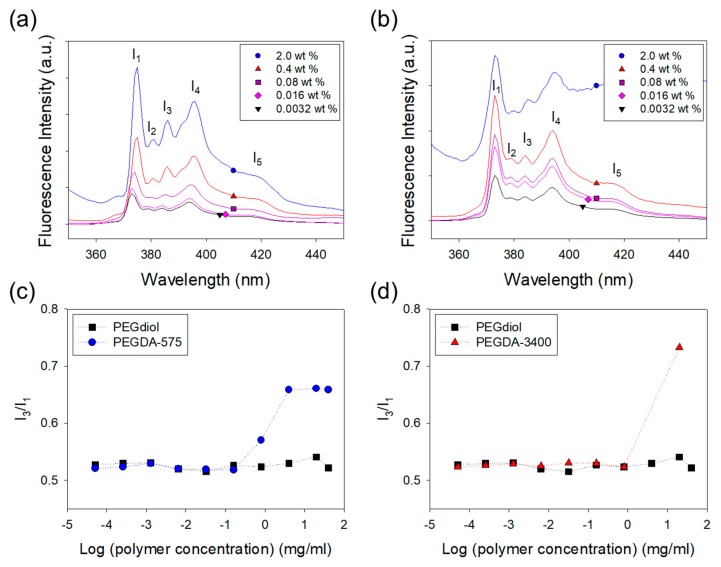
Fluorescence emission spectra of pyrene loaded in the (**a**) PEGDA-575 solution, (**b**) PEGDA-3400 solution were captured at various polymer concentrations. The ratio of third-to-first vibrational fine structure (*I_3_/I_1_*) in PEGDA increased while that of PEGdiol kept constant. The increase of *I_3_/I_1_* ratio of pyrene in the presence of polymer indicated that polymers formed aggregation. In (**c**) and (**d**), ● represents the solution of PEGDA-575, ▲ the solution of PEGDA-3400 and ■ the solution of PEGdiol.

**Figure 3 materials-12-01635-f003:**
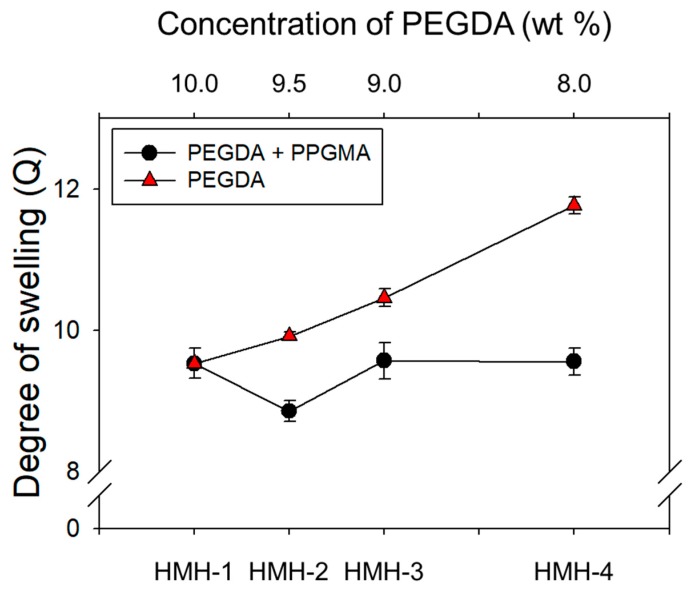
The degree of swelling (*Q*) of the hydrogel was calculated 10% PEGDA-575 hydrogels with decrease mass fraction of PEGDA (▲), and with increase mass fraction of PPGMA while keeping total polymer concentration constant (●). The hydrophobic chain in the hydrogel was the major cause of decreasing degree of swelling from 9.5 wt % to 10 wt %.

**Figure 4 materials-12-01635-f004:**
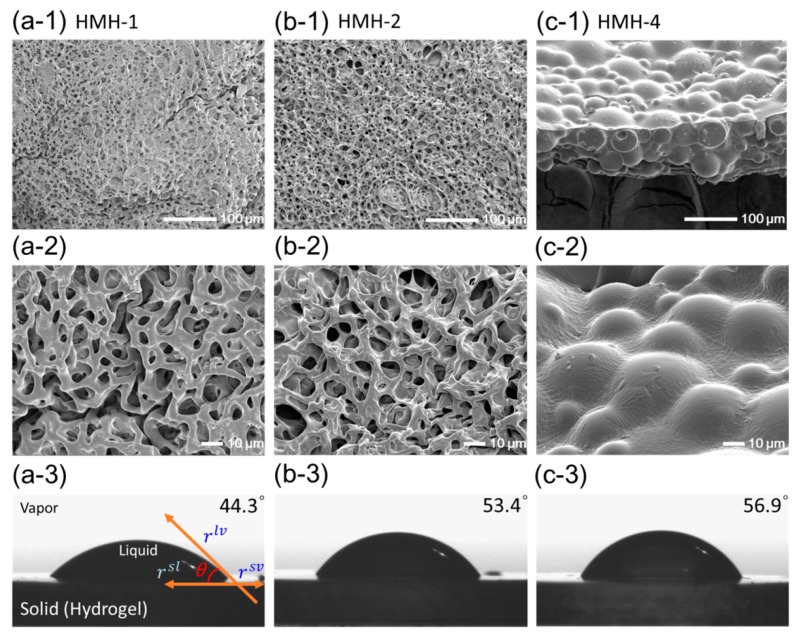
FE-Scanning electron microscope (FE-SEM) images and water contact angles of (**a**) pure PEGDA-575 hydrogel (HMH-1), hydrophobically-modified hydrogel, HMH-2 (**b**) at 0.5 wt % of PPGMAs and HMH-4 (**c**) at 2.0 wt % of PPGMAs.

**Figure 5 materials-12-01635-f005:**
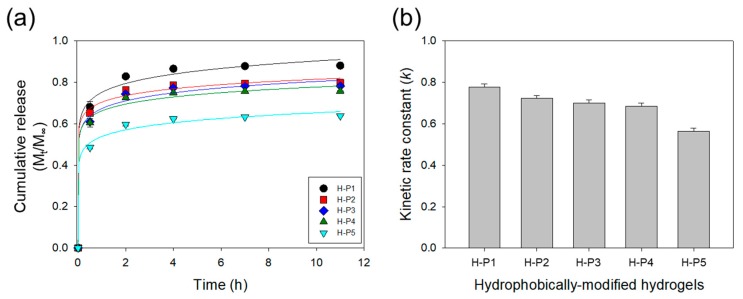
(**a**) Cumulative release profiles of bovine serum albumin (BSA) from the hydrogels. The solid lines represent the fitting curves used to quantify the kinetic rate constant (*k*) of release presented in (**b**).

**Table 1 materials-12-01635-t001:** Composition and characterization of the hydrophobically-modified hydrogel.

Sample	Composition		Theoretical Pore-Size (Å)	Experimental Degree of Swelling	Contact Angle (°)
	^2^PEGDA	^3^PPGMA			
**^1^HMH-1**	10.0	0.0	23.16	9.53 ± 0.21	44.30 ± 4.00
**HMH-2**	9.5	0.5	23.47	8.85 ± 0.15	53.40 ± 0.96
**HMH-3**	9.0	1.0	23.89	9.56 ± 0.26	54.84 ± 0.30
**HMH-4**	8.0	2.0	24.85	9.55 ± 0.19	56.87 ± 1.05

^1^ Hydrophobically modified hydrogel of Mn 575 g/mol (PEGDA-575); ^2^ Poly(ethylene glycol) diacrylate (wt %) and ^3^ Poly(propylene glycol) mathacrylate (wt %).

**Table 2 materials-12-01635-t002:** Composition of the hydrophobically-modified hydrogel to evaluate the protein release rate.

Sample	^3^H-P1	H-P2	H-P3	H-P4	H-P5
**^1^PEGDA**	20.0	19.85	19.7	19.5	18.0
**^2^PPGMA**	-	0.15	0.3	0.5	2.0

^1^ Poly(ethylene glycol) of Mw 3400 g/mol (PEGDA-3400, wt %); ^2^ Poly(propylene glycol) methacrylate (wt %) and ^3^ Hydrophobically-modified hydrogel to evaluate the protein release rate.

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
