# Peer review of "Tuning the Hydrophobicity of a Hydrogel Using Self-Assembled Domains of Polymer Cross-Linkers"

_materials, 2019, doi:10.3390/ma12101635_

Round 1
Reviewer 1 Report
This manuscript described an interesting method to incorporate hydrophobic PPGMA into a PEGDA. The experimental details were reported adequately. however, I have several comments.
the authors should provide a better introduction. What are the other efforts have been done to incorporate hydrophobic motifs into a hydrogel? Please provide references.
I am curious why do you choose PPGMA as a hydrophobic moiety?
Figure 1 (c), does the scheme really describes the self-assembly structure? I agree that the acrylate part is relatively hydrophobic. But does it really phase separated with PEG chain? doesn't PPGMA interact with the PEG chain? Please explain and provide references to this regard.
Please keep your naming system (including figures) consistent. I find it is confusing to read the figures (e.g. Figure 3, table1, figure 4). Figure 4 is very confusing!
water contact angle measurements are sensitive to surface roughness. Could this be the reason that you see the difference in WCA?
line 167-line 179: please illustrate more.
Author Response
Response to reviewer 1
This manuscript described an interesting method to incorporate hydrophobic PPGMA into a PEGDA. The experimental details were reported adequately. however, I have several comments.
1) The authors should provide a better introduction. What are the other efforts have been done to incorporate hydrophobic motifs into a hydrogel? Please provide references.
(Response) We appreciate the reviewer’s comment. We revised the part of introduction by including references, based on the reviewer’s suggestion, in the text follows;
In the text (page 1, line 32-36):
“In here, the hydrogels incorporated with hydrophobic motifs are required to recapitulate the hydrophobic/hydrophilic microenvironments in mineralized bone tissue [12, 13]. Extensive efforts have been made to control the hydrophobicity of hydrogels [13-15], and good examples include latex particles with coated hydrophobic segments to enhance the performance of hydrogels [15].”
2) I am curious why do you choose PPGMA as a hydrophobic moiety?
(Response) In this study, we used PPGMA as a model chain for hydrophobic moiety, because it has a similar molecular weight for that of PEGDA.
3) Figure 1 (c), does the scheme really describes the self-assembly structure? I agree that the acrylate part is relatively hydrophobic. But does it really phase separated with PEG chain? doesn't PPGMA interact with the PEG chain? Please explain and provide references to this regard.
(Response) The acrylates in the PEGDA (hydrophilic di-acrylates) is relatively hydrophobic as the reviewer mentioned, and therefore the PEGDA polymers could form self-assemblies in an aqueous solution, but they do not phase separation. Rather, the PPGMA used as hydrophobic molecules would experience the phase separation if they are incorporated in pre-polymer solution over the capacity of hydrophobic environments of PEGDA self-assemblies. We provide the reference concerning this regard as follows;
Jeong, J.H.; J. Schmidt, J.; Cha C.; Kong, H. Tuning responsiveness and structural integrity of pH responsive hydrogel using a poly(ethylene glycol) cross-linker. Soft Matter 2010, 6, 3930-3938.
Abdurrahmanoglu, S.; Can, V.; Okay, O. Polymer 2019, 50, 5449
4) Please keep your naming system (including figures) consistent. I find it is confusing to read the figures (e.g. Figure 3, table1, figure 4). Figure 4 is very confusing!
(Response) We have revised Figure 3 and 4 to keep the naming system consistent, related with the Table 1.
5) Water contact angle measurements are sensitive to surface roughness. Could this be the reason that you see the difference in WCA?
(Response) Thank you for your valuable comments. In this study, we have prepared the hydrogel in the same manner, including gelling environment, temperature, time, and plates. Therefore, we believe that the surface roughness of the hydrogel would not show significant differences, however, we agree that it need to report the effect of water contact angle on the surface roughness in the follow-up study.
6) Line 167-line 179: please illustrate more.
(Response) As the reviewer suggested, we have revised the manuscript to discuss it more clearly, as shown below:
In the text (page 6, line 191-203):
“The inner micro-structures of hydrophobically-modified hydrogels (HMHs) were examined with freeze-dried gels using SEM. Figure 4 shows the images of hydrogels depending on the introduced amounts of PPGMAs. Incorporation of the hydrophobic PPGMAs into the hydrogel made minimal difference in the 3D networked microstructures as compared to the pure PEGDA hydrogel (HMH-1) (Figure 4a & 4b). Therefore, the pore-size of the hydrogel was minimally affected by incorporating the hydrophobic chains. However, the hydrogel with 2.0 wt% of PPGMAs (HMH-4) exhibited relatively large hydrophobic micro-domains (Figure 4c-1). This result implicates that hydrophilic PEGDA and hydrophobic PPGMA molecules were separated into a two phases in the pre-gelled solution. As confirmed by the fluorescence assay, the PEGDAs, building blocks of a hydrogel are self-assembled with hydrophobic cores which can be used as micro-domains for incorporating PPGMA. However, incorporating of PPGMAs over a certain amount which is likely the loading capacity of the self-assemblies of PEGDAs resulted in the phase separation. Accordingly, the water contact angles (θw) measured at the surface of hydrogel increased with increasing of incorporating PPGMAs. But, the increase in the water contact angle has limited in HMH-4, due to the irregular arrangements of hydrophobic chains (Figure 4c-3). As a result, the hydrophobicity of a hydrogel could be tuned in a controlled manner by incorporating hydrophobic molecules using self-assembled domains of polymer cross-linkers.”

Reviewer 2 Report
The manuscript entitled: “Tuning the hydrophobicity of a hydrogel by self-assembly of polymer cross-linkers” submitted to Materials as Article describes synthesis and characterization of new modified hydrogels obtained in photopolymerization and containing hydrophobic domains. By different content of hydrophobic monomer in polymerization reaction the membrane morphology was tuned. The obtained hydrogels were characterized in the means of photoluminescence, degree of swelling, inner microstructure and release experiment. Authors showed in theirs work how the composition of pre-polymerization mixture influences the microstructure of hydrogels. Moreover, release of bovine serum albumin experiments were performed to confirm its applicability for possible medical use.
The topic presented in the manuscript discuss important aspects of tuneable synthesis of hydrogels with different morphologies is an interesting topic due to many possible applications. In my opinion described studies of tuneable hydrogels could be of interest for scientific community. However due to some mayor issues, I do not recommend this article for publishing in its current form.
Here is the list of issues that requires authors attention:
1. In general, whole manuscript requires a minor English revision, mainly spelling issues.
2. Introduction, generally: In my opinion the introduction lacks wider overview of current state of the art in the topic for this kind of materials.
3. Materials and Methods, generally: in my opinion Fig. 2 and Fig. 3 should be placed in Results and Discussion section with proper analysis.
4. Materials and Methods, generally: I missed in the manuscript description of experimental part information about the composition of stock solution, BSA encapsulation procedure, release experiment details (see Langmuir 2013, 29, 1601), measurements of polymer density and contact angle experiment (see Polymer 2016, 92, 50-57).
5. Results and Discussion, Fig 4: I assume that Authors obtained SEM imaging only for the surface of the hydrogels. I missed in the manuscript images of hydrogel membrane cross-section. It is not clear whether the membrane have symmetric/asymmetric structure. Authors do not provide any evidence of inner morphology.
6. Time dependant water up take study: In my opinion, time dependant study of solution uptake test would be profitable and provide information about time required to fully hydrate dry hydrogel (see Polymers 2018, 10 (4), 416; Polymer 2016, 92, 50-57).
7. Conclusions, in general: In my opinion conclusions should highlight more the obtained results presented in the article giving some details, i.e. the best obtained values.
8. References, in general: Authors should provide references form Materials journal.
Author Response
Response to reviewer 2
The manuscript entitled: “Tuning the hydrophobicity of a hydrogel by self-assembly of polymer cross-linkers” submitted to Materials as Article describes synthesis and characterization of new modified hydrogels obtained in photopolymerization and containing hydrophobic domains. By different content of hydrophobic monomer in polymerization reaction the membrane morphology was tuned. The obtained hydrogels were characterized in the means of photoluminescence, degree of swelling, inner microstructure and release experiment. Authors showed in theirs work how the composition of pre-polymerization mixture influences the microstructure of hydrogels. Moreover, release of bovine serum albumin experiments were performed to confirm its applicability for possible medical use.
The topic presented in the manuscript discuss important aspects of tuneable synthesis of hydrogels with different morphologies is an interesting topic due to many possible applications. In my opinion described studies of tuneable hydrogels could be of interest for scientific community. However due to some mayor issues, I do not recommend this article for publishing in its current form.
Here is the list of issues that requires authors’ attention:
1) In general, whole manuscript requires a minor English revision, mainly spelling issues.
(Response) We appreciate the reviewer’s comment. We have carefully revised the manuscript, specifically for English spellings.
2) Introduction, generally: In my opinion the introduction lacks wider overview of current state of the art in the topic for this kind of materials.
(Response) We appreciate the reviewer’s comment. We submitted this manuscript to the special issue, ‘Growth and Design of Inorganic Crystal’, and therefore, we have revised the manuscript with more specific introduction by including references, based on the reviewer’s suggestion, in the text follows;
In the text (page 1, line 27-39):
“Hydrogels have been extensively studied for use in various biomedical applications including drug delivery, tissue engineering and recently, Bio-MEMS(bio-microelectromechanical system) [1-4]. The successful use of the hydrogels in these applications greatly relies on their physical and chemical properties for maximizing their functionality [5-9]. Recently, mineralized hydrogel systems are being increasingly studied to understand bio-mineralization processes related to the development, repair, regeneration and remodeling of bone tissue [10,11]. In here, the hydrogels incorporated with hydrophobic motifs are required to recapitulate the hydrophobic/hydrophilic microenvironments in mineralized bone tissue [12,13]. Extensive efforts have been made to control the hydrophobicity of hydrogels [13-15], and good examples include latex particles with coated hydrophobic segments to enhance the performance of hydrogels [15]. However, incorporating of hydrophobic chains into a hydrogel often results in irregular arrangement of the hydrophobic chains, and further phase separation into hydrophobic domains, but limited efforts have been made to resolve this challenge in the hydrophobically-modified hydrogel.”
3) Materials and Methods, generally: in my opinion Fig. 2 and Fig. 3 should be placed in Results and Discussion section with proper analysis.
(Response) As the reviewer suggested, we have placed Fig.2 and Fig.3 in the section of Results and Discussion.
4) Materials and Methods, generally: I missed in the manuscript description of experimental part information about the composition of stock solution, BSA encapsulation procedure, release experiment details (see Langmuir 2013, 29, 1601), measurements of polymer density and contact angle experiment (see Polymer 2016, 92, 50-57).
(Response) We appreciate the reviewer’s comment. As the reviewer suggested, we have revised the part of experiments by including the suggested references, as shown below:
In the text (page 4, line 117-130):
“The inner micro-structures of the hydrophobically-modified hydrogels were analyzed by SEM (FE-Scanning electron microscope, JEOL-7001F) with an acceleration voltage of 5 kV. The hydrogels were lyophilized and covered with a platinum (Pt) layer produced by a vapor deposition. Values of the water contact angle (θw) on the surfaces of the hydrogels were measured by depositing a drop of water (4.0 μL) under atmospheric condition with DSA100 (KRÜSS). The contact angle was measured immediately after placing the water drop on the hydrogel surface [19]. Also, bovine serum albumin (BSA, Sigma) was used as a model protein to evaluate the protein release rate from the hydrophobically-modified hydrogels [20]. The PEGDA and PPGMA stock solution were dissolved in DI water to prepare the pre-gelled solution at 10 wt%. The pre-gelled solution was mixed with BSA and photo-initiator, the mixture immediately cast between glass plates with spacers of 1 mm thickness, and exposed by UV lamp about 10 min. After 10 min, gel disks with a diameter of 8 mm were punched. Each hydrogels containing BSA was suspended in 400μL of Dulbecco’s phosphate buffered saline (PBS, biowest) and incubated at 37 ˚C. At a designated time point, each PBS immersed hydrogel containing BSA was removed. Then, hydrogels were filled fresh PBS again thereby keeping the sink condition.”
[19] Bogdanozicz, K.A.; Rapsilber, G.A.; Reina, J.A.; Giamberini, M. Liquid crystalline polymeric wires for elective proton transport, part 1: Wires preparation. Polymer 2016, 92, 50-57.
[20] Bogdanozicz, K.A.; Tylkowski, B.; Giamberini, M. Preparation and characterization of light-sensitive microcapsules based on a liquid crystalline polyester. Langmuir 2013, 29, 1601-1608.
5) Results and Discussion, Fig 4: I assume that Authors obtained SEM imaging only for the surface of the hydrogels. I missed in the manuscript images of hydrogel membrane cross-section. It is not clear whether the membrane have symmetric/asymmetric structure. Authors do not provide any evidence of inner morphology.
(Response) We appreciate the reviewer’s comment. We have replaced the SEM image, in order to show the surface and cross-section of hydrogel, in Figure 4c-1.
In the Figure 4:
6) Time dependent water up take study: In my opinion, time dependent study of solution uptake test would be profitable and provide information about time required to fully hydrate dry hydrogel (see Polymers 2018, 10 (4), 416; Polymer 2016, 92, 50-57).
(Response) We appreciate the reviewer’s comment. As the reviewer suggested, we have provided the time required to fully hydrate it, in the text as follows.
In the text (page 3, line 101-103):
“The swelling ratio of the hydrogel at equilibrium was determined by measuring the weight of the hydrated gel and that of the dried gel. The hydrogels were immersed in DI water at room temperature and for 24 h to ensure the hydrogels were fully hydrated [18].”
[18] Bogdanozicz, K.A.; Pirone, D.; Judit, P.R.; Ambrogi, V.; Reina, J.A.; Giamberini, M. In situ raman spectroscopy as a tool for structural insight into cation non-ionomeric polymer interactions during ion transport. Polymers 2018, 10, 416-428.
7) Conclusions, in general: In my opinion conclusions should highlight more the obtained results presented in the article giving some details, i.e. the best obtained values.
(Response) As the reviewer suggested, we have revised the part of conclusions more clearly.
8) References, in general: Authors should provide references form Materials journal.
(Response) Thanks for the reviewer’s comment. We have carefully revised it, as follows;
In the part of references:
[4]. Moreno-Arotzena, O.; Meier, J.G.; Del Amo, C.; García-Aznar, J.M. Characterization of Fibrin and Collagen Gels for Engineering Wound Healing Models. Materials 2015, 8(4), 1636-1651.
[9]. Wang, Y.; Ma, M.; Wang, J.; Zhang, W.; Lu, W.; Gao, Y.; Zhang, B.; Guo, Y. Development of a Photo-Crosslinking, Biodegradable GelMA/PEGDA Hydrogel for Guided Bone Regeneration Materials. Materials 2018, 11(8), 1345, 1-12.

Reviewer 3 Report
The author reports a new method to incorporate hydrophobic domains regularly in a hydrogel by self-assembling of polymer cross-linkers in order to prepare materials with applications in drug delivery, cell therapies and tissue engineering.
In my opinion, some revisions are necessary:
1. In the Abstract, the authors used the word "motifs" too often. I suggest them to use, time to time, a synonym;
2. At page 2, Figure 1(b): In the right side picture, the authors should to put somehow in evidence the hydrophobic acrylate groups after polymerization. In the presented form, seems to disappear.
3. At page 2, line 51: "Schematic description of hydrogel forming form polymer". How is correct: form or from?
4. At page 2, lines 59-60: The authors should explain why they used two types of PEGDA (PEGDA-575 and PEGDA-3400);
5. At page 3, lines 75-76: the word "in" should be added "...was dissolved in DMSO...";
6. At page 5, line 147 (Table 1): Which type of PEGDA was used (575 or 3400)?
7. At page 6, line 172: It is about "Fig. 4a-3 & 4b-3"? It seems to be better Figs. 4c-1 and 4c-2;
8. At page 6, line 181 (Figure 4): Which type of PEGDA was used (575 or 3400)?
9. At page 6, line 183: Why the authors used PEGDA-3400 instead of PEGDA-575?
10. At page 7, line 197 (Figure 5): What represent the curves and bars labeled 1,2,3,4,5?
Author Response
Response to reviewer 3
The author reports a new method to incorporate hydrophobic domains regularly in a hydrogel by self-assembling of polymer cross-linkers in order to prepare materials with applications in drug delivery, cell therapies and tissue engineering.
In my opinion, some revisions are necessary:
1) In the Abstract, the authors used the word "motifs" too often. I suggest them to use, time to time, a synonym;
(Response) Thanks for the reviewer’s comment. We have used motifs, chains, and molecules in the text, as the reviewer suggested.
2) At page 2, Figure 1(b): In the right side picture, the authors should to put somehow in evidence the hydrophobic acrylate groups after polymerization. In the presented form, seems to disappear.
(Response) We appreciate the reviewer’s comment. Actually, Figure 1(b) shows the hydrogel forming from polymer cross-linkers in a pre-gelled solution in general. A polymer cross-linker, building block of a hydrogel has two or more acrylate groups which are relatively hydrophobic, compared to hydrophilic backbone polymer. Then, the polymer cross-linkers would be self-assembled with hydrophobic cores which can be used as micro-domains for incorporating hydrophobic molecules regularly. In order to explain this strategy, Figure 1(c) has been prepared in parallel.
3) At page 2, line 51: "Schematic description of hydrogel forming form polymer". How is correct: form or from?
(Response) Thanks for the reviewer’s comment. ‘from’ is correct, so we have replaced it.
4) At page 2, lines 59-60: The authors should explain why they used two types of PEGDA (PEGDA-575 and PEGDA-3400);
(Response) In this study, we have used PEGDA molecule as a model polymer cross-linker, but specifically PEGDA-3400 for the BSA release experiment. We have carefully indicated them in the text.
5) At page 3, lines 75-76: the word "in" should be added "...was dissolved in DMSO...";
(Response) As the reviewer suggested, we have corrected the part, as shown below:
In the text (page 3, lines 84-85):
“The photo-initiator, 2-hydroxy-4’-(2-hydroxyethoxy)-2-methylpropiophenone (Irgacure 2959, Sigma) was dissolved in DMSO at 10 wt% stock solution and added to the 1mL pre-gelled solution to form 0.2 wt% as the final concentration.”
6) At page 5, line 147 (Table 1): Which type of PEGDA was used (575 or 3400)?
(Response) We appreciate the reviewer’s comment. We have carefully revised the text including Table 1, by indicating PEGDA’s molecular weight appropriately.
Table 1:
Table 1. Composition and characterization of the hydrophobically-modified hydrogel.
Composition | Theoretical pore-size ( ) | Experimental degree of swelling | Contact angle (°) | ||
2PEGDA | 3PPGMA | ||||
1HMH-1 | 10.0 | 0.0 | 23.16 | 9.53 ± 0.21 | 44.30 ± 4.00 |
HMH-2 | 9.5 | 0.5 | 23.47 | 8.85 ± 0.15 | 53.40 ± 0.96 |
HMH-3 | 9.0 | 1.0 | 23.89 | 9.56 ± 0.26 | 54.84 ± 0.30 |
HMH-4 | 8.0 | 2.0 | 24.85 | 9.55 ± 0.19 | 56.87 ± 1.05 |
1 Hydrophobically modified hydrogel of Mn 575 g/mol (PEGDA-575).
2 Poly(ethylene glycol) diacrylate (wt%).
2 Poly(propylene glycol) methacrylate (wt%).
7) At page 6, line 172: It is about "Fig. 4a-3 & 4b-3"? It seems to be better Figs. 4c-1 and 4c-2;
(Response) We agree the reviewer’s comment. Therefore, we have replaced it, as the reviewer suggested.
8) At page 6, line 181 (Figure 4): Which type of PEGDA was used (575 or 3400)?
(Response) In Figure 4, PEGDA-575 was used, and then we have revised the Figure caption. Thanks for the reviewer’s comment.
9) At page 6, line 183: Why the authors used PEGDA-3400 instead of PEGDA-575?
(Response) We appreciate the reviewer’s comment. In this study, bovine serum albumin (BSA) was used as a model protein to evaluate the protein release rate from the hydrophobically-modified hydrogels. According to results of theoretical pore-size of hydrophobically-modified hydrogel (Table 1), PEGDA-575 hydrogel would be not suitable for this experiment, due to the dimension of BSA known as 140×40×40 Å[Ref]. Therefore, PEGDA-3400 was used to evaluate the protein release rate, instead of PEGDA-575.
[Ref] Wright, A.K.; Thompson, M.R. Hydrodynamic structure of bovine serum albumin determined by transient electric birefringence. Biophys J. 1975, 15(2 Pt 1), 137-141.
10) At page 7, line 197 (Figure 5): What represent the curves and bars labeled 1,2,3,4,5?
(Response) We have included the Table 2 for providing the composition of PEGDA-3400 hydrogels used to evaluate the protein release rate, and revised the labels of curves and bars in Figure 5.
Table 2:
Table 2. Composition of the hydrophobically-modified hydrogel to evaluate the protein release rate.
1H-P1 | H-P2 | H-P3 | H-P4 | H-P5 | |
1PEGDA | 20.0 | 19.85 | 19.7 | 19.5 | 18.0 |
2PPGMA | - | 0.15 | 0.3 | 0.5 | 2.0 |
1 Poly(ethylene glycol) of Mw 3400 g/mol (PEGDA-3400, wt%).
2 Poly(propylene glycol) methacrylate (wt%).
3 Hydrophobically-modified hydrogel to evaluate the protein release rate.
Figure 5:
Figure 5. (a) Cumulative release profiles of BSA from the hydrogels. The solid lines represent the fitting curves used to quantify the kinetic rate constant (k) of release presented in (b).

Reviewer 4 Report
The article deals with improvement of physically crosslinked network by the poly(propylene glycol) methacrylate (PPGMA). The article is quite interesting and novel. The reinforcement of the physical crosslink is very interesting.
However, I must recommend major revision
1. The Figure 1 (b) shows a modification of physical crosslink. However, there is not given evidence, that the chain of PPGMA is incorporated like in Figure 1. The fluorescence really shows higher degree of self-assembly, however, it is only indirect evidence.
2. The text of results can be restructured more logically.
3. I am lost in the identification of samples.
a. The Figure 2 identifies the samples by concentration in mg/ml,
b. Figure 3,4 +Table 1 identifies the concentration in wt%
c. Figure 5 numbers from 1-5.
4. What is the meaning of theoretical pore size? It seems that theoretical pore size of different samples is similar, although different swelling ratio. What you conclude from it? By the way, why the experiment of swelling were not provided with PEGDA-3400?
5. There are mixed three different factors in the text: 1. swelling (Figure 3, Table 1), 2. contact angle (Figure 4), and 3. release (Figure 5). The different factors describe different property of the hydrogel. They must be clearly distinguished.
6. I would expect the discussion in the text. The main topic of the article is self-assembly of physical crosslinks. What is relation of swelling, contact angle and release to the self-assembly of the physical crosslinks?
Minor revisions
1. The Figure 1a and b shows also chemical (covalent) crosslinking. The covalent crosslinking is mentioned only in methods. Why there is not mentioned in the introduction. The results are not clear, what was measured before and after chemical crosslinking.
2. The Fig. 2 is placed illogically at page 3, whereas the text description and commentary is at the page 5. As well the Fig. 3 and Table 1.
3. The word “Figure” is sometimes shortened (Fig…) and sometimes with full length (Figure …).
Final comments
I would recommend to change the title of paper. The self-assembly has been proven only indirectly. And the other properties do not have clear relation to self-assembly of physical crosslinks, because they were measured after chemical crosslinking (as I understood from text).
Author Response
Response to reviewer 4
The article deals with improvement of physically crosslinked network by the poly(propylene glycol) methacrylate (PPGMA). The article is quite interesting and novel. The reinforcement of the physical crosslink is very interesting.
However, I must recommend major revision
1. The Figure 1 (b) shows a modification of physical crosslink. However, there is not given evidence, that the chain of PPGMA is incorporated like in Figure 1. The fluorescence really shows higher degree of self-assembly, however, it is only indirect evidence.
(Response) We appreciate the reviewer’s comment. Actually, Figure 1(b) shows the hydrogel forming from polymer cross-linkers in a pre-gelled solution in general. A polymer cross-linker, building block of a hydrogel has two or more acrylate groups which are relatively hydrophobic, compared to hydrophilic backbone polymer. Then, the polymer cross-linkers would be self-assembled with hydrophobic cores which can be used as micro-domains for incorporating hydrophobic molecules regularly. In order to explain this strategy, Figure 1(c) has been prepared in parallel.
Yes, we agree the reviewer that the fluorescence result is an indirect evidence for the self-assemblies of PEGDAs. With this fluorescence assay, the other results including the water contact angle, SEM images, and swelling ratio would be helpful to ensure that the PPGMAs are incorporated in hydrogel.
2. The text of results can be restructured more logically.
(Response) As the reviewer suggested, we have carefully revised the manuscript to discuss it more clearly and logically.
3. I am lost in the identification of samples.
a. The Figure 2 identifies the samples by concentration in mg/ml,
b. Figure 3,4 +Table 1 identifies the concentration in wt%
c. Figure 5 numbers from 1-5.
(Response) We appreciate the reviewer’s comment. As the reviewer suggested, we have revised the manuscript to address this comment, as shown below:
In the Figure 2 (We have corrected the unit, ‘wt%’ instead of ‘mg/ml’ in Figure 2, as follows.):
In the Table 2 (We have included the Table 2 for providing the composition of PEGDA-3400 hydrogels used to evaluate the protein release rate, and revised the labels of curves and bars in Figure 5.):
Table 2. Composition of the hydrophobically-modified hydrogel to evaluate the protein release rate.
1H-P1 | H-P2 | H-P3 | H-P4 | H-P5 | |
1PEGDA | 20.0 | 19.85 | 19.7 | 19.5 | 18.0 |
2PPGMA | - | 0.15 | 0.3 | 0.5 | 2.0 |
1 Poly(ethylene glycol) of Mw 3400 g/mol (PEGDA-3400, wt%).
2 Poly(propylene glycol) methacrylate (wt%).
3 Hydrophobically-modified hydrogel to evaluate the protein release rate.
In the Figure 5:
4. What is the meaning of theoretical pore size? It seems that theoretical pore size of different samples is similar, although different swelling ratio. What you conclude from it? By the way, why the experiment of swelling were not provided with PEGDA-3400?
(Response) In this study, we have used PEGDA molecule as a model polymer cross-linker, but specifically PEGDA-3400 for the BSA release experiment. We have carefully revised and indicated them in the text.
The theoretical pore size was evaluated from the polymer volume fraction and the unperturbed mean-square end-to-end distance of the monomer unit, based on the PEGDA concentration. Therefore, as the reviewer indicated, the theoretical pore size showed a minimal difference in each condition, but slightly it was increased. However, interestingly, in case of the hydrophobically-modified hydrogels, the Q (degree of swelling ratio) of hydrogel was decreased with increasing PPGMA portion from 0 to 0.5 wt%, despite of the decrease of PEGDA concentration in Figure 3. These results indicated that PPGMA regulates the degree of swelling of the hydrogel at a mass fraction of less than 0.5 wt% of PPGMA as a hydrophobic repulsion. In contrast, above 0.5 wt% of the mass fraction of PPGMAs, the decreased mass fraction of PEGDAs has a predominant role to the Q of hydrogel (That’s why the PEGDA-3400 was not used for this experiment). Note that PPGMAs are not related to regulate the ξ of hydrogel, because PPGMA molecule has a single acrylate group which is not acting as a cross-linker. However, the ξ of hydrogel from 0 to 0.5 wt% of PPGMAs was decreased despite of the decrease of PEGDAs’ concentration. As addressed above, this result is attributed to hydrophobic repulsion by PPGMA.
We appreciate the reviewer’s comments. Therefore, we have revised the manuscript to address this part more carefully.
5. There are mixed three different factors in the text: 1. swelling (Figure 3, Table 1), 2. contact angle (Figure 4), and 3. release (Figure 5). The different factors describe different property of the hydrogel. They must be clearly distinguished.
(Response) We appreciate the reviewer’s comment. Yes, we agree that they are different factors for describing a hydrogel’s property. Therefore, we have revised the part of results to mention clearly why each factor was evaluated to show the effect of incorporated hydrophobic chains in a hydrogel.
6. I would expect the discussion in the text. The main topic of the article is self-assembly of physical crosslinks. What is relation of swelling, contact angle and release to the self-assembly of the physical crosslinks?
(Response) We appreciate the reviewer’s comment.
But, this study is not much related on the physical crosslinks (We think we are not able to understand well the reviewer’s wording, physical crosslinks). Actually, this study presents a strategy to incorporate hydrophobic domains regularly in a hydrogel using self-assembled domains formed with polymer cross-linkers, building blocks of a hydrogel. (In general, introduction of hydrophobic molecules into a hydrogel often results in irregular arrangement of the motifs, and further phase separation of hydrophobic domains.) A polymer cross-linker, building block of a hydrogel has two or more acrylate groups which are relatively hydrophobic, compared to hydrophilic backbone polymer. Then, the polymer cross-linkers would be self-assembled with hydrophobic cores which can be used as micro-domains for incorporating hydrophobic molecules regularly.
Minor revisions
1. The Figure 1a and b shows also chemical (covalent) crosslinking. The covalent crosslinking is mentioned only in methods. Why there is not mentioned in the introduction. The results are not clear, what was measured before and after chemical crosslinking.
(Response) In this study, we are not focusing on the physical cross-linking, but chemical cross-linking, therefore, we mentioned the polymer cross-linkers in the part of introduction. The polymer cross-linkers are the building blocks of a hydrogel, but they are formed as self-assemblies in the pre-gelled solution, so we were trying to use it for incorporating hydrophobic chains into a hydrogel.
2. The Fig. 2 is placed illogically at page 3, whereas the text description and commentary is at the page 5. As well the Fig. 3 and Table 1.
(Response) As the reviewer suggested, we have placed Fig.2 and Fig.3 in the section of Results and Discussion.
3. The word “Figure” is sometimes shortened (Fig…) and sometimes with full length (Figure …).
(Response) We appreciate the reviewer’s comment. We have corrected it, with ‘Figure’ (full length) in the text.
Final comments
I would recommend to change the title of paper. The self-assembly has been proven only indirectly. And the other properties do not have clear relation to self-assembly of physical crosslinks, because they were measured after chemical crosslinking (as I understood from text).
(Response) Thanks for the reviewer’s comment. In order to avoid any misunderstanding, as the reviewer’s comment, we have changed the title, as follows:
“Tuning the hydrophobicity of a hydrogel using self-assembled domains of polymer cross-linkers”

Round 2
Reviewer 1 Report
The revised manuscript improved significantly. The revised manuscript improved significantly. The experimental details were well presented and statements were well-supported.
One minor comment: please incorporate your response for my comment#3 into the manuscript. It would be helpful for readers to better understand your rationale.
Author Response
Response to reviewer 1
The revised manuscript improved significantly. The revised manuscript improved significantly. The experimental details were well presented and statements were well-supported.
One minor comment: please incorporate your response for my comment#3 into the manuscript. It would be helpful for readers to better understand your rationale.
(Response) We appreciate the reviewer’s comment. We have revised the part of introduction, based on the reviewer’s suggestion, in the text follows;
In the text (page 1-2, line 42-45):
“A polymer cross-linker, building block of a hydrogel has two or more acrylate groups which are relatively hydrophobic, compared to hydrophilic backbone polymer. Then, the polymer cross-linkers would be self-assembled with hydrophobic cores which can be used as micro-domains for incorporating hydrophobic molecules regularly (Figure 1(b)).”
Reviewer 2 Report
The manuscript entitled: “Tuning the hydrophobicity of a hydrogel by self-assembly of polymer cross-linkers” submitted to Materials as Article for second revision. The manuscript contains description of synthesis and characterization of new modified hydrogels containing hydrophobic domains, obtained in photopolymerization.
After the revision of the corrected manuscript and the response of Authors to the comments form the first revision, in my opinion manuscript shows significant improvement and seems suitable to be published in Materials, hence I recommend this manuscript for publishing.
I would only suggest correction of two things:
1. I would suggest to replace symbol & for word “and”
2. I would spell check the name of authors in references, i.e. “Bogdanozicz” should be corrected for “Bogdanowicz”.
Author Response
Response to reviewer 2
The manuscript entitled: “Tuning the hydrophobicity of a hydrogel by self-assembly of polymer cross-linkers” submitted to Materials as Article for second revision. The manuscript contains description of synthesis and characterization of new modified hydrogels containing hydrophobic domains, obtained in photo-polymerization.
After the revision of the corrected manuscript and the response of Authors to the comments from the first revision, in my opinion manuscript shows significant improvement and seems suitable to be published in Materials, hence I recommend this manuscript for publishing.
I would only suggest correction of two things:
1) I would suggest to replace symbol & for word “and”
(Response) We appreciate the reviewer’s comment. We have replaced the symbol from ‘&’ to ‘and’ in the text;
In the text (page 6, line 191-195):
“~ as compared to the pure PEGDA hydrogel (HMH-1) (Figure 4a and 4b). Therefore, the pore-size of the hydrogel was minimally affected by incorporating the hydrophobic chains. However, the hydrogel with 2.0 wt% of PPGMAs (HMH-4) exhibited relatively large hydrophobic micro-domains (Figure 4c-1 and 4c-2).”
2) I would spell check the name of authors in references, i.e. “Bogdanozicz” should be corrected for “Bogdanowicz”.
(Response) We appreciate the reviewer’s comment. We have checked and corrected the name of author in the reference.;
In the text (page 7, line 297-305):
18. Bogdanowicz, K.A.; Pirone, D.; Judit, P.R.; Ambrogi, V.; Reina, J.A.; Giamberini, M. In situ raman spectroscopy as a tool for structural insight into cation non-ionomeric polymer interactions during ion transport. Polymers 2018, 10, 416-428.
20. Bogdanowicz, K.A.; Rapsilber, G.A.; Reina, J.A.; Giamberini, M. Liquid crystalline polymeric wires for elective proton transport, part 1: Wires preparation. Polymer 2016, 92, 50-57.
21. Bogdanowicz, K.A.; Tylkowski, B.; Giamberini, M. Preparation and characterization of light-sensitive microcapsules based on a liquid crystalline polyester. Langmuir 2013, 29, 1601-1608.